# Cancer Therapy-Related Cardiac Dysfunction in Patients Treated with a Combination of an Immune Checkpoint Inhibitor and Doxorubicin

**DOI:** 10.3390/cancers14092320

**Published:** 2022-05-07

**Authors:** Seon-Hwa Lee, Iksung Cho, Seng-Chan You, Min-Jae Cha, Jee-Suk Chang, William D. Kim, Kyu-yong Go, Dae-Young Kim, Jiwon Seo, Chi-Young Shim, Geu-Ru Hong, Seok-Min Kang, Jong-Won Ha, Sun-Young Rha, Hyo-Song Kim

**Affiliations:** 1Division of Cardiology, Severance Cardiovascular Hospital, Yonsei University College of Medicine, Seoul 03722, Korea; seonhwa@yuhs.ac (S.-H.L.); ivorykyu@yuhs.ac (K.-y.G.); entialov@yuhs.ac (D.-Y.K.); eprangiana@yuhs.ac (J.S.); cysprs@yuhs.ac (C.-Y.S.); grhong@yuhs.ac (G.-R.H.); smkang@yuhs.ac (S.-M.K.); jwha@yuhs.ac (J.-W.H.); 2Department of Biomedical Systems Informatics, Yonsei University College of Medicine, Seoul 03722, Korea; chandryou@yuhs.ac; 3Department of Radiology, Chung-Ang University Hospital, Chung-Ang University College of Medicine, Seoul 03722, Korea; minjaecha@cau.ac.kr; 4Department of Radiation Oncology, Yonsei University College of Medicine, Seoul 03722, Korea; changjeesuk@yuhs.ac; 5Chung-Ang University College of Medicine, Seoul 03722, Korea; wdk95344@gmail.com; 6Division of Oncology, Yonsei University College of Medicine, Seoul 03722, Korea; rha7655@yuhs.ac

**Keywords:** immune checkpoint inhibitor, cancer therapy-related cardiac dysfunction, doxorubicin, sarcoma

## Abstract

**Simple Summary:**

Immune checkpoint inhibitors (ICIs) have demonstrated promising efficacy against various types of cancer. Although ICIs exhibit strong anti-cancer effects, they can cause adverse cardiac events. However, only a few case reports have focused on whether ICIs and cardiotoxic agents increase the risk of cancer therapy-related cardiac dysfunction (CTRCD). Therefore, we evaluated cardiac dysfunction in patients with sarcoma who were receiving doxorubicin with or without ICI using echocardiography and left ventricular global longitudinal strain (LVGLS). This study shows that ICIs may increase the risk of CTRCD when used concomitantly with cardiotoxic agents. Furthermore, the serum troponin-T level was significantly elevated in patients receiving doxorubicin with ICIs. Therefore, CTRCD should be monitored in patients who are being treated with ICIs by cardiac biomarkers and echocardiography including myocardial strain.

**Abstract:**

Backgrounds: There are scarce data on whether immune checkpoint inhibitors (ICIs) increase the risk of cardiac dysfunction when used with cardiotoxic agents. Thus, we evaluated cardiac dysfunction in patients with sarcoma receiving doxorubicin with or without ICI using echocardiography and left ventricular global longitudinal strain (LVGLS). Methods: A total of 95 patients were included in this study. Echocardiography and LVGLS were evaluated at baseline and follow-up (at 3 and 6 months of chemotherapy) and compared with the doxorubicin (Dox; *n* = 73) and concomitant ICI with doxorubicin (Dox-ICI; *n* = 22) groups. Cancer therapy-related cardiac dysfunction (CTRCD) was defined as a left ventricular ejection fraction (LVEF) drop of >10% and LVEF of <50% (definite CTRCD), LVEF drop of >10%, LVEF of ≥50%, and LVGLS relative reduction of >15% (probable CTRCD) at six months. Results: There were no significant differences in age, cumulative dose of doxorubicin, and cardiovascular risk factors between the two groups. At baseline, the LVEF was similar in the Dox and Dox-ICI groups (*p* = 0.493). In the Dox group, LVEF decreased to 59 ± 6% (Δ −7 ± 1.3%, *p* < 0.001) and LVGLS decreased from −17.3 ± 3.2% to −15.4 ± 3.2% (Δ −10.1 ± −1.9%, *p* < 0.001) at six months. In the Dox-ICI group, LVEF decreased to 55 ± 9% (Δ −9 ± 2.1%, *p* < 0.001), along with a significant decrease in LVGLS (from −18.6 ± 1.9% to −15.3 ± 3.6%, Δ −12.4 ± −2.4%, *p* < 0.001). Over a median follow-up of 192 days, there were no cases with clinical manifestations of fulminant myocarditis. In the Dox group, definite and probable CTRCD were observed in seven (10.1%) and five (7.4%) patients, respectively. In the Dox-ICI group, definite and probable CTRCD were observed in four (19%) and four (19%) patients, respectively. The total number of patients who developed CTRCD was significantly higher in the Dox-ICI group than in the Dox group (38.1% vs. 17.4%, *p* = 0.042). Serum troponin-T level was significantly higher in the Dox-ICI group than in the Dox group (53.3 vs. 27.5 pg/mL, *p* = 0.023). Conclusions: ICIs may increase the risk of CTRCD when used with cardiotoxic agents. CTRCD should be monitored in patients treated with ICIs by cardiac biomarkers and echocardiography, including LV-GLS.

## 1. Introduction

Soft tissue sarcoma (STS) comprises diverse histological subtypes with distinct clinical and molecular features. Despite the heterogeneity, patients with advanced STS are generally prescribed the same treatment, including doxorubicin- or ifosfamide-based regimens [1,2]. As a long-standing standard treatment, the recommended maximum lifetime cumulative dose of doxorubicin is 400–450 mg/m^2^, and up to 50% of patients have been observed to have asymptomatic deterioration of the left ventricular ejection fraction (LVEF) [3].

In addition to cytotoxic chemotherapy, immune checkpoint inhibitors (ICIs) have demonstrated promising efficacy against various STS subtypes. For advanced sarcomas, monotherapy with pembrolizumab, an anti-PD-1 antibody, was associated with clinically meaningful efficacy [4]. Nivolumab, another anti-PD-1 monoclonal antibody, in combination with the cytotoxic T-lymphocyte-associated protein 4 (CTLA-4) inhibitor ipilimumab, exhibited therapeutic activity; the combination received regulatory approval as a standard treatment for refractory STS [5]. Treatment of doxorubicin in combination with pembrolizumab showed favorable outcomes in STS [6]. Durvalumab is another monoclonal antibody that blocks PD-L1 binding to PD-1 and CD80. Although ICIs exhibit strong anti-cancer effects, they can cause a range of immune-related adverse events (irAEs), mostly involving the skin, liver, lung, endocrine system, and gastrointestinal tract. Fulminant myocarditis is the most recognized cardiac irAE. ICIs-associated myocarditis occurs early during ICI treatment. Previous retrospective studies have estimated the incidence of ICI-associated myocarditis to be between 0.27% [7] and 1.14% [8]. Clinical presentation of myocarditis is very heterogeneous and may include different symptoms, such as chest pain, dyspnea, pulmonary edema, arrhythmias, and even cardiogenic shock [9]. About half of the patients with ICI-associated myocarditis have no evidence of reduced ejection fraction. On the other hand, serum troponin is elevated in the vast majority of cases. ICI-associated myocarditis can be difficult to diagnose given the multiple tests needed to exclude other diagnoses. The multimodalities for diagnosing ICI-associated myocarditis are biopsy (gold standard), MRI, elevated biomarker, and echocardiography. Management of ICI associated myocarditis consists of holding ICI and administering immunosuppression agents, such as steroids, or a second line of immunosuppression agents in refractory cases. Surveillance for ICI-related myocarditis is important due to its potential lethality. According to the recently proposed protocol, surveillance includes regular testing of the troponin levels. In patients with elevated troponin values on surveillance, further workup such as ECG and echocardiography should be performed [10]. Besides ICI-associated myocarditis, pericardial disease, arrhythmias, conduction abnormalities, left ventricular dysfunction, and Takotsubo-like cardiomyopathy have also been reported [11,12]. A recent study showed that patients treated with ICI therapy had increased rates of cardiac events; moreover, the absolute risk was higher than that reported in previous pharmacovigilance studies [13]. Additionally, ICIs have been tested in combination with conventional cytotoxic and targeted therapies [14].

However, there are only a few case reports on whether ICIs and cardiotoxic agents may increase the risk of cardiac dysfunction [15]. Possible predictors of cardiotoxicity include preexisting cardiovascular risk factors, cumulative dose, and age, which were suggested for chemotherapy or ICI monotherapy, although their clinical significance is uncertain [16,17]. Therefore, we aimed to evaluate the cardiotoxicity of ICIs in patients with sarcoma receiving doxorubicin with or without ICI using comprehensive echocardiographic parameters, left ventricular global longitudinal strain (LVGLS) and cardiac biomarkers.

## 2. Methods

### 2.1. Study Population

Patients with histologically confirmed metastatic and/or recurrent sarcoma between December 2019 and December 2020 were enrolled in a prospective clinical trial (ClinicalTrials.gov Identifier: NCT03802071) according to the following criteria (Dox-ICI cohort): (1) age ≥19 years, (2) LVEF at baseline echocardiography ≥50%, and (3) without prior exposure to ICIs and anthracycline. Briefly, patients received concomitant ICI (durvalumab, 1500 mg) with doxorubicin (75 mg/m^2^) every three weeks for 6 cycles followed by maintenance treatment with ICI (durvalumab) every three weeks. The dose of doxorubicin was reduced if there were any side effects.

Additionally, we retrospectively analyzed 78 patients with sarcoma who received doxorubicin treatment from February 2014 to June 2020 (Dox cohort) and had a baseline LVEF of 50% or higher. The echocardiography was followed at 3 and 6 months after treatment. In total, 108 patients with sarcoma were included in this study. Demographic data and information of the study participants were collected from the electronic medical records. The study was approved by the institutional review board, and a prospective cohort of patients provided written informed consent before enrolment in accordance with the Declaration of Helsinki and the Guidelines for Good Clinical Practice (IRB No. 4-2021-0638). However, the need for informed consent was waived for the retrospectively analyzed patients.

### 2.2. 2D and Speckle Tracking Echocardiography

Echocardiography was performed at the initiation of chemotherapy and after three and six months of chemotherapy using a standard ultrasound machine (Vivid E9; GE Medical Systems, Wauwatosa, WI, USA; Philips iE33; Philips Healthcare, Best, The Netherlands) with a 2.5–3.5 MHz probe. Standard echocardiographic measurements were performed according to the recommendations of the American Society of Echocardiography [18]. LVEF was measured using the biplane method of disks (modified Simpson’s rule) from apical four- and two-chamber views. The left atrial volume index was assessed manually using Simpson’s method at the end of the ventricular systole and indexed to the body surface area [18]. Systolic (S’) and peak early (e’) and late (A’) diastolic annular velocities were obtained via tissue Doppler imaging (TDI) of the septal mitral annulus. Pulmonary artery systolic pressure was estimated using the following formula: 4 × (tricuspid regurgitant velocity [m/s])^2^ + right atrial pressure (mmHg). From two-dimensional images of the apical two-, three-, and four-chamber views, LVGLS was measured offline using a vendor-independent software package (TomTec software; Image Arena 4.6, Munich, Germany) [19] (Appendix A). Echocardiographic data and strain values were analyzed by an experienced cardiologist blinded to the clinical data of the patients.

### 2.3. Echocardiography-Based Definition of Cancer Therapy-Related Cardiac Dysfunction

In this study, cancer therapy-related cardiac dysfunction (CTRCD) was defined as a decline in LVEF by >10 absolute percentage points to a value <50% [20] by echocardiography at six months of chemotherapy. We defined these patients as definite CTRCD in this study. In subclinical cardiotoxicity, probable cardiotoxicity was defined as a decline in LVEF by >10 absolute percentage points to a value ≥50% accompanied by a relative reduction in LVGLS >15% [20]. These patients were defined as having probable CTRCD in the present study.

### 2.4. Statistical Analysis

Continuous variables are presented as mean ± standard deviation (SD) or median (interquartile range) and were compared using the paired Student’s t-test (for normally distributed data) or the Mann–Whitney U test (for non-normally distributed data). Categorical variables were presented as absolute numbers and percentages and were analyzed using the chi-square or Fisher’s exact test. Propensity score matching analysis was performed while adjusting for clinically confounding factors. Confounding factors included age, sex, and Adriamycin cumulative dose. Patients were matched in a 1:1 analysis. After the matching process, two groups were re-evaluated for their baseline characteristics and echocardiographic parameters. Statistical significance was defined as a two-sided *p*-value < 0.05. All analyses were performed using IBM SPSS Statistics for Windows (version 25.0; IBM Corp., Armonk, NY, USA).

## 3. Results

### 3.1. Patients and Demographics

A total of 30 patients in the Dox-ICI cohort and 78 patients in the Dox cohort were enrolled between September 2014 and December 2020. Moreover, 13 patients were excluded due to discontinuation of therapy (*n* = 5), loss to follow-up (*n* = 3), and poor quality of images (*n* = 5), and the remaining 95 patients were included in the final analysis (CONSORT, Figure 1). The baseline demographic and oncologic characteristics according to the cancer therapy are described in Table 1. There were no significant differences with respect to age, cardioprotective agents such as beta-blockers, angiotensin-converting enzyme inhibitors (ACEIs), and angiotensin receptor blockers (ARBs). No significant difference in cardiovascular risk factors except chronic kidney disease was observed between the two groups. Moreover, the cumulative dose of doxorubicin was not significantly different between the Dox cohort and the Dox-ICI cohort (400 ± 56 vs. 395 ± 49 mg/m^2^, *p* = 0.820). After propensity score matching, there was no significant difference in the baseline demographic.

### 3.2. Echocardiographic Characteristics: Baseline and Serial Changes

At baseline, there were no significant differences in LVEF between the Dox cohort and the Dox-ICI cohort (66 ± 6 vs. 65 ± 5%, *p* = 0.493, Table 2). Additionally, the initial LVGLS was not significantly different between the two groups. At 3-month follow-up, LVEF was preserved in the Dox-ICI cohort, whereas it was significantly decreased from 66.6 ± 6.3% to 61.6 ± 9.2% within the normal range in the Dox cohort. However, LVGLS was not significantly different at 3-month follow-up (Appendix A).

In the 6-month follow-up echocardiography, LVEF along with LVGLS were significantly reduced compared with the baseline echocardiography in both groups (Figure 2A,B). However, there was no significant difference in the absolute change in LVEF and relative change in LV-GLS in both groups. Further, the tissue Doppler parameters, S’ and e’ velocity were significantly reduced in the 6-month follow-up echocardiography in the Dox-ICI cohort, but not in the Dox cohort (Figure 2C,D). There was no significant difference in results before and after matching analysis.

### 3.3. Cancer Therapy-Related Cardiac Dysfunction and Clinical Outcomes

With a median follow-up of 192 days (range, 169–210 days), a total of 20 patients developed CTRCD, among which 11 patients (55%) developed definite CTRCD and 9 (45%) developed probable CTRCD. At the time of occurrence of CTRCD, the median (IQR) of LVEF and LVGLS among the patients was 48% (31–58%) and −10.8% (−7.0 to −12.6%).

In the Dox cohort, definite and probable CTRCD were observed in seven (10.1%) and five (7.4%) patients, respectively, whereas in the Dox-ICI cohort, it was observed in four (19%) and four (19%) patients, respectively. The total number of patients who developed CTRCD was significantly higher in the Dox-ICI cohort than in the Dox cohort (38.0% vs. 17.4%, *p* = 0.042) (Figure 3). Among those with definite CTRCD, 11 (100%) patients showed symptoms of heart failure, and 10 (91%) of those discontinued chemotherapy. In the definite CTRCD of the Dox group, one out of seven patients developed definite CTRCD before six cycles of doxorubicin. Six out of seven patients developed definite CTRCD after six cycles of doxorubicin. These patients discontinued doxorubicin and received guideline-directed medical treatment for heart failure; they showed improvement in cardiac function with LVEF being increased to >50%. In the definite CTRCD of the Dox-ICI group, four patients developed definite CTRCD after six cycles of doxorubicin and durvalumab. These patients also discontinued doxorubicin and durvalumab and received medical treatment for heart failure according to symptoms and laboratory results. Three out of four had follow-up images, whereas the remaining one was lost to follow-up due to cancer progression. All three patients showed improvement in cardiac function (with LVEF ≥50%) 6 months to 1 year after treatment of heart failure.

Out of total patients, all-cause mortality occurred in one patient in the Dox-ICI group and three patients in the Dox group. The cause of all-cause mortality was cancer progression. When analyzing cardiovascular events in patients with definite and probable CTRCD, there was no cardiovascular death and hemodynamically significant heart block in the Dox group and Dox-ICI group. Cardiogenic shock occurred in two patients (25.0%) in the Dox-ICI group and two patients (16.7%) in the Dox group. Furthermore, there was no fatal or fulminant myocarditis or irreversible heart failure in the Dox-ICI group.

### 3.4. Predictors of Cancer Therapy-Related Cardiac Dysfunction

Next, we identified potential predictors of CTRCD using clinical and biomarker parameters. At baseline, hypertension and dyslipidemia were more prevalent in patients who developed CTRCD than in those who did not (Table 3). Regarding treatment-related factors, concomitant use of ICI was significantly more prevalent in the CTRCD than in the non-CTRCD (40.0% vs. 18.7%, *p* = 0.044). Serum troponin-T as a cardiac biomarker was obtained at the time of follow-up echocardiography in only patients who developed CTRCD (*n* = 20). More than half of the patients who developed probable CTRCD and definite CTRCD had elevated troponin-T levels above the upper normal limit in both cohorts (Appendix A). In patients with definite CTRCD, serum troponin-T was significantly more elevated in the Dox-ICI cohort (*n* = 4) than in the Dox cohort (*n* = 7) (69.0 vs. 29.7 pg/mL, *p* = 0.012). Consistently, in all patients who developed CTRCD, serum troponin-T was significantly more elevated in the Dox-ICI cohort (*n* = 8) than in the Dox cohort (*n* = 12) (53.3 vs. 27.5 pg/mL, *p* = 0.023) (Figure 4).

## 4. Discussion

In the current study, we evaluated CTRCD by adding LVGLS to conventional echocardiographic parameters. We observed that concomitant use of ICI with doxorubicin increased the risk of occurrence of CTRCD at 6-month follow-up. Furthermore, in patients with CTRCD, serum troponin-T was elevated, which was significantly increased in the ICI and Dox group compared to the Dox group. This study also showed the clinical course of CTRCD by regular echocardiographic monitoring of troponin-T, and suggests the proper treatment option for those cases.

In previous studies, ICI-related cardiotoxicity is well known as myocarditis which results from suppression of immune regression [7]. Although ICI-related myocarditis has been rarely reported, fulminant myocarditis is fatal. ICI-related myocarditis had the highest fatality rate among all the fatal cases secondary to ICI toxicity [21]. A multicenter registry found that the prevalence of myocarditis was 1.14%, with a median time to onset of 34 days after starting ICI therapy [8]. In another study, myocarditis was reported to occur after a median period of 65 days from the initiation of ICI therapy [12]. Despite these differences in the timing of occurrence, it is evident that most ICI-related myocarditis occurs early during treatment. More recent studies reported that cancer patients treated with ICI had an increased incidence of atherosclerotic cardiovascular events, including myocardial infarction, ischemic stroke, and coronary revascularization, compared to non-ICI-treated cancer patients, mediated by the accelerated progression of atherosclerosis [22]. Among lung cancer and melanoma patients treated with ICIs, the absolute risk of cardiac events was 7–10% per year, which is higher than reported by a previous nationwide pharmacovigilance Danish study [13]. Several previous studies were focused on ICI-related cardiovascular events, but there are little data on whether ICI increases cardiac dysfunction, particularly when used with cardiotoxic agents.

To the best of our knowledge, this is the first study to evaluate the frequency, characteristics and clinical course of ICI-related cardiac dysfunction when combined with the cardiotoxic agent doxorubicin. Our study showed that ICIs also increase the prevalence of cardiac dysfunction without the occurrence of acute fulminant myocarditis.

In this study, we evaluated serial changes (baseline, and 3 and 6 months) in echocardiography after concomitant ICI and doxorubicin treatment. We evaluated CTRCD by analyzing LVEF, measured by 2D echocardiography, and LVGLS, measured by speckle tracking echocardiography, simultaneously. Although LVEF is a commonly used echocardiographic parameter, it is still limited to the detection of subclinical cardiac dysfunction and is dependent on loading conditions. LVGLS is a marker of the cardiac contractile function of the LV. Compared to LVEF, LVGLS is more sensitive for evaluating the systolic function of the LV and is considered a parameter for the detection of subclinical LV dysfunction arising from various etiologies [23,24]. Therefore, LVGLS is the most important parameter for the early detection of CTRCD [25]. Furthermore, the measurement of GLS after initiation of potentially cardiotoxic chemotherapy had a good prognostic performance for subsequent CTRCD [26].

Consistent with the definition given in a previous study [20,27,28], we defined definite and probable CTRCD based on reduction in LVEF and LVGLS from baseline values in the study population. There was no significant difference in the occurrence of definite CTRCD between the Dox-ICI cohort and the Dox cohort (19.0% vs. 10.1%; *p* = 0.256). However, while considering LVGLS and LVEF simultaneously in the evaluation of CTRCD, the occurrence of cardiac dysfunction including probable CTRCD after treatment was more frequent in the Dox-ICI cohort than in the Dox cohort. Furthermore, TDI showed a significant reduction of S’ and e’ in the Dox-ICI cohort after treatment [29]. TDI is more sensitive than conventional Doppler because it is less influenced by loading conditions and provides an improved evaluation of cardiac function changes during anthracycline therapy [29]. Interestingly, the serum high sensitive troponin-T level in all patients with CTRCD was significantly higher in the Dox-ICI group than in the Dox group, providing evidence for myocardial damage. High sensitive troponin-T is a specific biomarker of myocardial damage. Elevated high sensitive troponin-T levels have been reported to be associated with histological evidence of myocardial injury induced by doxorubicin [30]. Several studies have shown that troponin-T level has been utilized for the early detection of doxorubicin-induced cardiotoxicity in chemotherapy-treated patients [31,32].

In risk factors of ICI-related cardiac dysfunction, dual ICI therapy is the well-established risk factor for ICI-associated myocarditis [7]. Apart from dual ICI therapy, there is a lack of evidence of risk factors for ICI-related cardiotoxicity. A meta-analysis of cardiotoxicity associated with ICI therapy demonstrated that 40% of the patients who developed ICI cardiotoxicity had cardiovascular risk factors, such as hypertension [17]. In our study, cardiovascular risk factors, including hypertension and dyslipidemia, were significantly more prevalent in all patients with CTRCD. However, in all patients with CTRCD in the Dox-ICI cohort, there was no significant difference among cardiovascular risk factors. This discrepancy in results might be due to the relatively younger age of the population with lower frequency of cardiovascular risk factors in this study compared to the previous study. Thus, evaluation of cardiovascular risk factors in ICI-related cardiac dysfunction was difficult in this study.

A recent study showed that the prior use of doxorubicin was associated with higher mortality in ICI-treated patients [15]. However, it remains unclear whether the incidence of ICI-related cardiotoxicity is high when ICIs are combined with non-ICI agents for cancer treatment. In preliminary results for avelumab with axitinib (a VEGF inhibitor) [33], one patient (2%) developed fatal myocarditis. In the present study, CTRCD was more prevalent and the troponin-T level was elevated in patients of the Dox-ICI cohort compared to the Dox group. This result suggests the possible increased risk of cardiotoxicity when ICIs are combined with non-ICI cardiotoxic agents. However, during long-term follow-up, cardiac function was recovered after discontinuation of chemotherapy and treatment for heart failure, thus cardiotoxicity is considered as reversible. The mechanism underlying induction of cardiotoxicity when ICIs are combined with non-ICI cardiotoxic agents remains unclear; thus, further studies are warranted.

This study had some limitations. First, although LVEF is a commonly used echocardiographic parameter, it has many limitations in the detection of cardiac dysfunction, including myocarditis. Therefore, defining the CTRCD through only echocardiographic parameters in this study was a major limitation. However, to overcome such limitations, we performed an analysis of a myocardial strain to detect subclinical cardiac dysfunction. Second, it was a single-center study. Secondly, we acknowledge that the small number of patients reduces the statistical power of our result. Furthermore, considering the multiple numbers of analyses performed in this limited cohort, without the adjusting for confounding factors, requires that study results should be cautiously interpreted. However, we analyzed homogenous patients with sarcoma, therefore, tumor-related bias and age-related cardiotoxicity were less incorporated. Third, serum troponin-T level was measured at the time of CTRCD diagnosis but not at baseline or serial changes. Therefore, comparison with other factors, such as echocardiographic parameters in assessing the association of cardiac events, is limited. Since elevated troponin-T levels are also known to predict LV dysfunction in patients receiving cancer therapy [31], additional prospective analyses are required to confirm these findings.

## 5. Conclusions

In conclusion, ICIs may increase the risk of CTRCD, particularly when used concomitantly with cardiotoxic agents. Thus, it is important to regularly check cardiac biomarkers in patients who are treated with ICIs and perform additional echocardiography including strain analysis to detect and treat CTRCD at an early stage. Larger prospective and confirmatory studies are warranted.

## Figures and Tables

**Figure 1 cancers-14-02320-f001:**
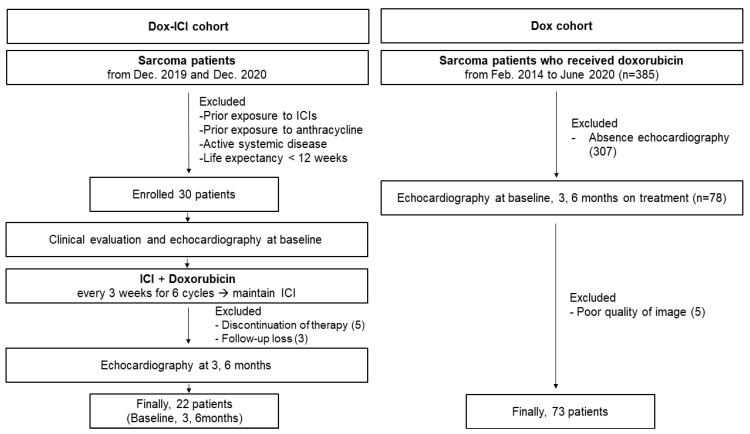
A graphical representation of the patient selection process.

**Figure 2 cancers-14-02320-f002:**
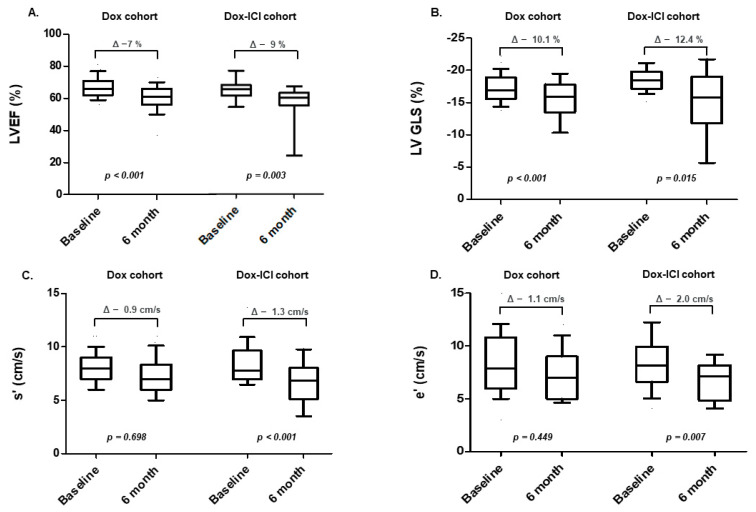
Changes in echocardiographic parameters after chemotherapy. (**A**,**B**) reveal that LVEF and LVGLS were significantly reduced after chemotherapy in both cohorts. (**C**,**D**) show that tissue Doppler imaging velocities (S’ and e’) were significantly decreased in the Dox-ICI cohort, but not in the Dox cohort.

**Figure 3 cancers-14-02320-f003:**
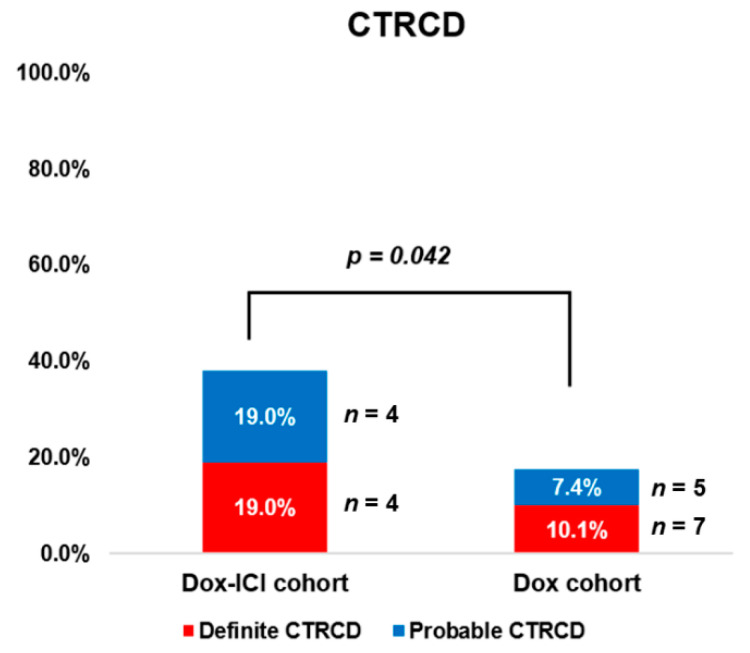
Higher incidence of CTRCD in the Dox-ICI cohort than in the Dox cohort.

**Figure 4 cancers-14-02320-f004:**
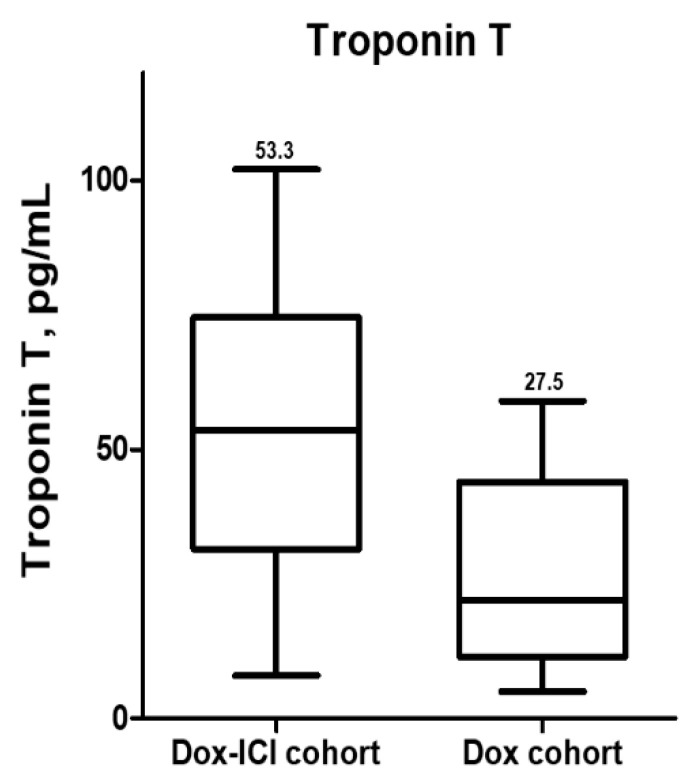
In patients with CTRCD, the level of troponin-T was significantly more elevated in the Dox-ICI cohort than in the Dox cohort.

**Table 1 cancers-14-02320-t001:** Baseline characteristics of the study population.

		Unmatching		Matching *	
	Dox-ICI Cohort (*n* = 22)	Dox Cohort (*n* = 73)	*p* Value	Dox Cohort (*n* = 22)	*p* Value †
Age	51 ± 14	53 ± 14	0.587	51 ± 14	0.985
Male, *n* (%)	14 (48.3)	27 (39.1)	0.402	14 (48.3)	1.000
Cardiovascular risk factors					
Hypertension, *n* (%)	5 (17.2)	16 (22.2)	0.598	6 (20.7)	0.783
Diabetes, *n* (%)	3 (10.3)	7 (9.5)	0.976	4 (13.8)	0.687
Dyslipidemia, *n* (%)	2 (6.9)	1 (1.3)	0.153	0 (0)	0.150
Atrial fibrillation, *n* (%)	1 (3.4)	5 (6.8)	0.474	3 (10.3)	0.300
Coronary artery disease, *n* (%)	1 (3.4)	1 (1.3)	0.523	0 (0)	0.313
Chronic kidney disease, *n* (%)	0 (0)	12 (16.4)	0.017	5 (17.2)	0.019
Baseline physical findings	117 ± 11	113 ± 19	0.409	113 ± 19	0.409
Systolic blood pressure, mmHg	117 ± 11	113 ± 19	0.409	114 ± 19	0.395
Diastolic blood pressure, mmHg	71 ± 10	70 ± 17	0.841	69 ± 21	0.705
BSA, kg/m^2^	1.67 ± 0.17	1.63 ± 0.22	0.472	1.63 ± 0.20	0.448
Medication					
Beta blocker, *n* (%)	1 (4.5)	2 (2.7)	0.667	1 (4.5)	0.879
ACE-I or ARB, *n* (%)	3 (13.6)	10 (13.6)	0.469	5 (22.7)	0.447
Statin, *n* (%)	5 (22.7)	12 (16.4)	0.145	6 (27.2)	0.256
Adriamycin cumulative dose, mg/m^2^	395 ± 49	400 ± 56	0.820	398 ± 55	0.954

ACE-I, angiotensin-converting enzyme inhibitor; ARB, angiotensin receptor blocker; BSA, body surface area. * Matching for age, sex and adriamycin cumulative dose; † *p* for Dox-ICI group and matching Dox group.

**Table 2 cancers-14-02320-t002:** Echocardiographic parameters at baseline and follow-up during chemotherapy.

				Unmatching			Matching *		
	Dox-ICI Cohort (*n* = 22)			Dox Cohort (*n* = 73)			Dox Cohort (*n* = 22)		
	Baseline	6-Month	*p* Value	Baseline	6-Month	*p* Value	Baseline	6-Month	*p* Value
LVEF, %	65 ± 5	56 ± 8	0.003	66 ± 6	59 ± 6	<0.001	66 ± 6	60 ± 9	<0.001
Change in LVEF, %	−9.3 ± 2.1			−7.0 ± 1.3		0.376	−6.9 ± 1.9		0.188 †
(Absolute change)									
LVEDD, mm	47 ± 3	45 ± 2	0.291	48 ± 3	48 ± 2	0.674	48 ± 3	48 ± 2	0.624
LVESD, mm	30 ± 2	31 ± 1	0.952	31 ± 3	32 ± 3	0.856	31 ± 3	33 ± 3	0.122
LAVI, mL/m^2^	27.6 ± 1.3	27.5 ± 2.1	0.965	27.8 ± 11.0	28.4 ± 12.0	0.776	27.8 ± 11.0	28.8 ± 7.2	0.650
E velocity, cm/s	0.68 ± 0.16	0.58 ± 0.28	0.127	0.69 ± 0.16	0.65 ± 0.20	0.196	0.69 ± 0.16	0.64 ± 0.18	0.095
A velocity, cm/s	0.63 ± 0.24	0.51 ± 0.29	0.051	0.64 ± 0.16	0.67 ± 0.17	0.332	0.64 ± 0.16	0.67 ± 0.17	0.148
S’, cm/s	8.42 ± 2.06	6.15 ± 3.19	<0.001	8.1 ± 1.5	7.2 ± 2.2	0.698	8.1 ± 1.5	7.2 ± 2.2	0.375
e’, cm/s	8.12 ± 2.39	6.07 ± 3.27	0.007	8.1 ± 2.7	7.0 ± 2.6	0.449	8.1 ± 2.7	7.1 ± 2.3	0.225
a’, cm/s	9.21 ± 3.30	6.89 ± 4.28	0.01	8.7 ± 1.8	8.1 ± 2.3	0.101	8.7 ± 1.8	8.1 ± 2.3	0.419
E/e’ ratio	8.76 ± 2.27	9.60 ± 2.27	0.602	9.1 ± 2.7	9.8 ± 4.1	0.209	9.1 ± 2.7	9.6 ± 2.9	0.589
PASP, mmHg	22 ± 9	27 ± 9	0.504	26 ± 7	28 ± 9	0.353	26 ± 7	28 ± 8	0.267
LVGLS, %	−18.6 ± 1.9	−15.3 ± 3.6	0.015	−17.3 ± 2.3	−15.4 ± 3.6	<0.001	−17.3 ± 2.3	−15.6 ± 2.8	<0.001
Change in LVGLS, %	−12.4 ± −2.4			−10.1 ± −1.9		0.392	−10.0 ± −1.8		0.296 †
(Relative change)									

* Matching for age, sex and adriamycin cumulative dose. † *p* for Dox-ICI group and matching Dox group.

**Table 3 cancers-14-02320-t003:** Comparison between CTRCD and non-CTRCD.

	CTRCD (*n* = 20)	Non-CTRCD (*n* = 75)	*p* Value
Cardiovascular risk factors			
Hypertension, *n* (%)	8 (40.0)	11 (15.3)	0.029
Diabetes, *n* (%)	3 (15.0)	8 (10.2)	0.703
Dyslipidemia, *n* (%)	3 (15.0)	1 (1.2)	0.034
Coronary artery disease, *n* (%)	1 (5.0)	1 (1.2)	0.397
Atrial fibrillation or atrial flutter, *n* (%)	2 (10.0)	4 (5.1)	0.611
Oncologic risk factor			
Adriamycin cumulative dose, mg/m^2^	430 ± 133	394 ± 113	0.084
Concomitant use of ICI	8 (40.0)	14 (18.7)	0.044
Baseline LV systolic function			
LVEF, %	67 ± 6	66 ± 6	0.408
LVGLS, %	−18.0 ± 1.9	−17.4 ± 3.2	0.296

## Data Availability

The datasets analyzed in this study are available from the corresponding author upon reasonable request.

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
