# Peer review of "Cancer Therapy-Related Cardiac Dysfunction in Patients Treated with a Combination of an Immune Checkpoint Inhibitor and Doxorubicin"

_cancers, 2022, doi:10.3390/cancers14092320_

Round 1
Reviewer 1 Report
Please, see attached my comments.

Author Response
"Please see the attachment."

Reviewer 2 Report
The article „Cancer therapy-related cardiac dysfunction in patients treated with a combination of an immune checkpoint inhibitor and doxorubicin“ by Lee et al. tackles a an interesting and up-to-date topic in cardio-oncology. In my opinion this article is of interest to a broad readership of oncologists and cardiologists.
The strength of the work is the prospective nature of the study and the solid documentation of echocardiographic results.
Nevertheless, there are some issues that need to be adressed before publication:
- The authors mainly focus on the GLS to detect cardiotoxicity. Since two different echo machines (two manufactures) were used, could the authors clarify how they measured GLS, since the manufactures‘ softwares show differences in accurancy and value distributions. Did the authors use a non-manufacture specific software for the analysis? If not, which patients were measured with which machine (Philips or GE)?
- Troponin values were seen higher in ICI and Dox treated patients with cardiotoxicity. Could the authors state on myocarditis in these patients (aparat from non fatal myocarditis cases)? Were biopsies or cardiac MRI performed in some patients?
- It would help to improve the paper, if biomarkers and echo parameters are compared directly, e.g. in multivariant logistic regression
- Are outcome data (all-cause mortality, cardiovascular death) available?
- The patients numbers vary a lot between Dox and ICI and Dox. I would recommend to perform propensity score matching to show, if there is still a difference between both groups after matching (in comparable sizes of the groups). The differences in LVEF and GLS are rather little between both groups. Maybe the biomarkers should be put more into the focus, even though values are not available in all patients.
Author Response
"Please see the attachment."

Round 2
Reviewer 1 Report
I appreciate the modifications that have been done. Please, see attached my slight edits on the sections you added (corrections highlighted in yellow). Thank you.

Author Response
"Please see the attachment."
Reviewer 2 Report
Thank you for revising the manuscript. It is much improved.
Author Response
Thank you very much for your review.